# The Plasma Membrane—An Integrating Compartment for Mechano-Signaling

**DOI:** 10.3390/plants9040505

**Published:** 2020-04-14

**Authors:** Frank Ackermann, Thomas Stanislas

**Affiliations:** Center for Plant Molecular Biology, University of Tuebingen, 72076 Tuebingen, Baden-Wuerttemberg, Germany; frank.ackermann@uni-tuebingen.de

**Keywords:** mechanical stress, Arabidopsis, plasma membrane, cell wall integrity, mechanosensitive signaling pathways, receptor-like kinase

## Abstract

Plants are able to sense their mechanical environment. This mechanical signal is used by the plant to determine its phenotypic features. This is true also at a smaller scale. Morphogenesis, both at the cell and tissue level, involves mechanical signals that influence specific patterns of gene expression and trigger signaling pathways. How a mechanical stress is perceived and how this signal is transduced into the cell remains a challenging question in the plant community. Among the structural components of plant cells, the plasma membrane has received very little attention. Yet, its position at the interface between the cell wall and the interior of the cell makes it a key factor at the nexus between biochemical and mechanical cues. So far, most of the key players that are described to perceive and maintain mechanical cell status and to respond to a mechanical stress are localized at or close to the plasma membrane. In this review, we will focus on the importance of the plasma membrane in mechano-sensing and try to illustrate how the composition of this dynamic compartment is involved in the regulatory processes of a cell to respond to mechanical stress.

## 1. Introduction

Plants are able to sense their mechanical environment. For example, leaves of touch-sensitive plants, such as Mimosa (*Mimosa pudica*) and Venus flytrap (*Dionaea muscipula*), in response to stimulation fold and bend or close to catch a prey, respectively [1]. Plant architecture is also affected by mechanical stimuli with plant stems becoming shorter and stiffer when exposed to wind, nicely indicating that plants are able to change their phenotypic features in response to their mechanical environment [2]. This is also true at a smaller scale for morphogenesis. The shape changes, both at the cell and tissue levels, rely on development-related gene expression and on the concentration and polarity of growth factors that induce major changes in the internal pressure and the property of extra-cellular matrix, which by definition depend on the laws of mechanics. Indeed, cell- and tissue-shape as well as internal (turgor) pressure determine the physical stress, thereby inducing a mechanical strain (deformation). The deformation of the tissue depends on its geometrical features and material properties, defining morphogenesis as the accumulation over time of the relative deformation of an object induced by stress pattern [3]. But mechanics is not just a read-out. Over the past years, it became more and more clear that this mechanical strain is a signal regulating growth and developmental pattern formation in plants and animals. Morphogenesis involves mechanical signals that influence specific patterns of gene expression and trigger cytoskeletal reorganizations [4].

Plant cell mechanics are defined by the balance between turgor pressure and cell wall stiffness. The turgor pressure is the hydrostatic pressure of the cytoplasm exerted to the apoplasm. At the cellular level, turgor pressure pushes the plasma membrane against the cell wall and causes in-plane mechanical tension within the cell [5]. This osmotic pressure can be regulated by synthesizing osmolytes or by exchanging osmolytes with the apoplastic compartment, a strategy that is also used when plants are exposed to salt, draught, freezing or heat stress [6]. The exact relationship between cell deformation, turgor pressure, and osmolyte exchange still remains poorly understood. A recent study showed that turgor pressure is heterogeneous in between cells of a same tissue and anticorrelates with the number of neighboring cells. This result, combined with physical models of pressurized cells, suggest that water conductivity may contribute to growth control [7]. Indeed, if the generated pressure is above the yielding pressure threshold of the cell wall, turgor pressure induces a plastic deformation of the cell [8]. Cell deformation can be also regulated by the modification of the cell wall. The cell wall is made of highly organized cellulose microfibrils connected by hemicellulose and embedded in a matrix of pectin and structural proteins. Cellulose microfibrils are the major load-bearing component of the wall, and their orientation therefore largely dictates the anisotropic cell mechanics [9]. Each cellulose microfibril is composed of β-1,4-linked glucan chains synthesized at the cell surface by cellulose synthase (CESA) complexes, which are highly mobile proteins localized at the plasma membrane [10].

Among all structural components of plant cells that are involved in the response to mechanical stress and cell plasticity, the plasma membrane has received very little attention. So far in *Arabidopsis*, key players, that are able to sense and modify the turgor pressure and the mechanical property of the cell wall, have in common that they are localized at or neighboring the plasma membrane and that this localization is crucial for their function. In this review, we will focus on the importance of the plasma membrane in mechano-sensing and the key players that are involved in this dynamic network. Our work is based on the latest contributions on the field and summarizes what is known, but also tries to tackle some questions that still need to be answered.

## 2. The Plasma Membrane—A Signaling Compartment

The principal components of the plasma membrane are lipids (phospholipids and sterols), proteins, and carbohydrate groups that are attached to some of the lipids and proteins. For a long time, the plasma membrane was considered as a simple barrier delimiting the intracellular space from the external environment [11]. This vision has evolved into a more complex notion where proteins in the plasma membrane help the cell to interact with its environment [12]. For example, plasma membrane proteins carry out functions as diverse as transporting nutrients across the plasma membrane, receiving chemical signals from outside the cell, translating chemical signals into intracellular actions, and sometimes anchoring the cell to intra- or extracellular matrices [13]. It has become clear that the huge diversity of lipids that constitute the plasma membrane are not only structural components but play a crucial role in signaling. Indeed, low abundance lipids such as phosphatidic acid, phosphatidylserine or phosphoinositide phosphates are known as signaling lipids, as of their rapid turnover it is easy to significantly modify their concentration. Their production is controlled by lipid-modifying enzymes, for example the glycerophospholipids are modified by phospholipases, lipid kinases, and/or phosphatases [14]. Stress-activation of these enzymes will modify the concentration and the spatial accumulation of a signaling lipid into the tissues and/or the cell. This will generate secondary messenger, but also modulate the physical properties of membranes, such as the membrane tension, fluidity, curvature, asymmetry, surface charges, or clustering of proteins [15]. Over the past 30 years, research carried out by biologists, biophysicists, and biochemists has revealed the existence of particular domains within the plasma membrane called membrane rafts, that are liquid-ordered domains more tightly packed than the surrounding non-raft phase of the bilayer [16]. In plants, the existence of membrane rafts was initially suggested by the identification of a low-density TritonX-100 insoluble fraction [11]. Possible artifacts generated by the use of detergents at low temperatures keep the membrane raft concept under active debate. Finally, in situ characterization of plasma membrane organization by immunoelectron microscopy and super-resolution microscopy revealed the presence of <100 nm clusters on the PM of Arabidopsis (review in [11]). These membrane rafts are plasma membrane platforms in which proteins involved in signaling can selectively interact with effector molecules [11]. For example, there are predictions, based on analysis of *Arabidopsis* mutants, that membrane rafts influence cellulose biosynthesis by regulating the localization, the structural integrity or the activity of the cellulose synthase complex [17].

This new vision of the plasma membrane as a dynamic signaling compartment pave the way to integrate it in cell mechanics pathways and to understand how the plasma membrane tunes the perception and transduction of cell mechanics.

## 3. The Plasma Membrane—A Sensor of the Mechanical State of the Cell

In order to maintain the cell mechanic equilibrium between the turgor pressure and the resistance of the cell wall, the cell has to be able to sense the state of these two parameters. 

The cell wall integrity pathway describes molecular components that sense cell wall integrity including signals arising in the wall, sensors that detect changes at the cell surface and secondary effects induced by a modification of the cell wall [13]. In *Arabidopsis*, changes in wall stress can be directly sensed by receptor-like kinases (RLK) which generally consist of a central transmembrane domain, with an extracellular domain to sense the cell wall, and an intracellular kinase domain to relay signals to downstream components. They form, with over 600 members, one of the biggest gene families in *Arabidopsis*. THESEUS1 (THE1) and FERONIA (FER) are two *Catharanthus roseus* RECEPTOR-LIKE KINASE1-LIKE (CrRLK1L) receptors that gained a lot of attention recently [18]. The extracellular domains of both receptors show sequence homology to the animal malectin carbohydrate-binding domains, which led to speculations that both are able to interact with cell-wall polysaccharides and can potentially serve as cell wall sensors [18]. Whereas no carbohydrate components of the wall have been identified as ligands that bind THE1, it was recently shown that FER is able in vitro to bind demethylesterified homogalacturonan, a major structural domain of pectin [19]. Plasma membrane localization of FER was described to depend on a family of lipid-modified proteins called glycosylphosphatidylinositol-anchored proteins (GPI-AP) [20]. GPI-APs are a class of membrane proteins containing a soluble protein attached by a glycolipid anchor to the external leaflet of the plasma membrane. A preformed GPI anchor is synthetized and attached to the C-terminus of the nascent protein in the endoplasmic reticulum [21]. It was shown that FER and the GPI-AP called LRE-like GPI-AP1 (LLG1) interact with each other in the endoplasmic reticulum, where proteins are assembled before both proteins are moved via the Golgi to the plasma membrane and are located into membrane domains [20]. Interestingly, THE1 and FER act also as receptors for rapid alkalinization factor (RALF) peptides that are trapped by non-covalent bounds to the cell wall. Cell wall stress or strain may release these peptides and the perception of these peptides by THE1 and FER may be transduced into a mechanical signal [22]. Other cell wall sensors are also located at the plasma membrane. Wall-associated kinases possess an extracellular domain capable of binding the pectin backbone and pectin fragments in the cell wall and leucine-rich repeat receptor-like kinases such as FEI1 and FEI2 are required for cellulose synthesis and anisotropic cell expansion [23,24]. Future challenges will be to understand the molecular mechanisms that allow receptors located at the plasma membrane to sense the state of the cell wall and to transduce an adapted signal to respond accordingly to mechanical stresses.

Lipids are also potential direct sensors of the cell wall. Glycosyl Inositol Phospho Ceramides (GIPCs) are structured in a trihydroxylated LCB moiety amidified by a 2-hydroxylated Very Long Chain Fatty Acid linked to an inositol–glucuronic acid and can be further modified by adding different saccharides to the polar head. It was shown that GIPCs are capable of binding rhamnogalacturonan II, which is a fraction of pectin [25]. GIPCs are lipids with very long saturated acyl chains organized in membrane domains at the plasma membrane [26]. It was calculated that the carbon atom acyl chains of GIPC are too long to fit in one leaflet and will therefor penetrate within the inner leaflet and interdigitate with the inner leaflet lipid acyl chains [26]. This makes GIPCs putative links between the plasma membrane and the cell wall, which could access the plasma membrane to recognize the current physical state of the cell.

So far, no turgor pressure rheostat has clearly been characterized in *Arabidopsis*. There is evidence suggesting that WALL-ASSOCIATED-KINASES may provide a molecular mechanism linking cell wall sensing and turgor maintenance in growing cells [27]. In vitro, the WAK extracellular domain binds to pectin oligosaccharides in a Ca^2+^-dependent manner [28] and single *wak2* mutation exhibits a dependence on appropriate sugars and salts and reduces vacuolar invertase activity in the roots [27], which in turn is known to be involved in systems of turgor maintenance. Yet there is, for now, no proof for this hypothesis. By contrast, sensing the state of the turgor has intensively been studied in yeast. In yeast, osmosensing mechanisms involves a two-component signal transducer and a MAP kinase cascade. The transmembrane histidine kinase Synthetic lethal of N-end rule (Sln1), which has an extracellular sensor domain and cytoplasmic kinase and receiver domain, is such a signal transducer. Activation of *Sln1* mediates a multistep phosphotransfer reaction (phosphorelay), leading to the activation of MAPK cascades [29]. The plasma membrane-localized ARABIDOPSIS HISTIDINE KINASEs (AHK) 1 was shown to be able to rescue yeast strains deficient for SLN1, suggesting that AHK1 can function in principle as a turgor sensing component [30]. Unfortunately, there is so far no support for this function *in planta*. Interestingly, AHK1 exhibits a low similarity compared to other histidine kinases found in plants, while there is a significant structural similarity to Sln1. Analysis of protein sequences of histidine kinases using bioinformatics tools also predicted the presence of membrane-spanning region or trans-membrane domains. AHK1 has two hydrophobic regions in its N-terminal half [31,32]. These hydrophobic regions indicate that the AHK1 is capable of recognizing amphipathic phospholipids, which might regulate its localization. Yet, the exact mechanism for plasma membrane localization is not fully understood.

When pathogens breach the cell wall to access the cytoplasm of plant cells during the penetration phase, oligosaccharide fragments from the cell wall are released and recognized as plant damage-associated molecular patterns that activate plant defense mechanisms [33]. This will also modify the strength of the cell wall and increase the turgor-induced tension at the plasma membrane that could potentially be a signal to activate plant defense. Indeed, it is already well known in bacteria and yeast that the activation of osmosensors is correlated with thickness-induced conformational changes in the transmembrane domains of those sensors. For example, antimicrobial peptides cause thinning effects of the plasma membrane leading to changes of the transmembrane domain of PhoQ, a bacterial histidine kinase, resulting in the phosphorylation and subsequent activation of the kinase (reviewed in [34]). So far, there is no similar mechanism described in plants.

Finally, the most conceptually straightforward membrane-bound mechanosensory principle comes from the direct effect of anisotropic forces of the lipid bilayer and their changes that can reshape embedded proteins, especially channels that can open in response to a mechanical stress [35]. Mechanosensitive channels represent the best example of coupling protein conformations to the mechanics of the surrounding cell membrane. 

## 4. Mechanosensitive Channels at the Plasma Membrane

The cellular uptake of osmolytes is an essential signaling pathway in all living organisms, as this changes the concentration of signaling molecules inside and outside the cell. Increasing or decreasing solute concentration into the cell will directly modify the turgor pressure [36]. On the other way, activation of H^+^-ATPases leads to increased extracellular pH and wall loosening and internalization of calcium will activate respiratory burst oxidase homologues (RBOH) that will produce reactive oxygen species in the apoplast such as H_2_O_2_. This will activate peroxidases inducing cell wall stiffening or OH^-^ that will interfere with xyloglucan and induce cell wall loosening [37]. In all cases, this will modify the cell mechanics by interfering in the equilibrium of turgor pressure and resistance to the cell wall. The activity of such channels can also be altered by physical forces and are called mechanosensitive ion channels. Indeed, as a subtype of mechanical forces, osmotic stress generates membrane tension, which applies forces to the embedded channels. Mechanosensitive ion channels are integral membrane proteins that form aqueous ion pores across the plasma membrane. Patch clamp is a powerful technique that allows in a single cell the recording of channel activity with a high resolution in time and ion flux. Using this technique, it was shown that the state of the ion channel depends on the tension force that is applied to the plasma membrane [38,39]. No membrane tension leads to closed channels. In the presence of tension conformational changes lead to the opening (activation), thereby allowing ions to flow through the membrane into the cell, generating electrical currents. Releasing the tension again leads to the closer and thereby deactivation of the channel [38,39].

Small Conductance Mechanosensitive Ion Channel (MscS)-Like (MSL) proteins, first identified by homology to the bacterial mechanosensitive channel MscS from *Escherichia coli*, are non-selective and stretch-activated channels [40]. These channels generate, based on the tension of the plasma membrane, a 1-nanoSiemen (nS) conductance, thereby regulating different signaling cascades. The primary function of the bacterial MscS is to regulate the response to hypoosmotic stress, by releasing osmolytes from the bacterium [41]. Ten MscS homologs exist In *Arabidopsis*. Three of those homologs are localized to organellar membranes and seven are predicted to localize to the plasma- and vacuolar-membranes. Studies in *Arabidopsis* root cell protoplasts provide evidence that the homologs MSL9 and MSL10, both localized at the plasma membrane, are genetically required for the predominant mechanosensitive channel activity. These studies support the hypothesis that MSL proteins are mechanosensitive receptors [42,43,44].

Calcium (Ca^2+^) signaling is an important signaling pathway, taking place in different compartments of the cell. In this review, we will focus on the importance of the plasma membrane for proper calcium signaling in response to a mechanical stress. In most organisms, calcium ions act as primary regulators of the initial responses to osmotic pressure with a rapid increase in the cytosolic free Ca^2+^ concentration [45]. The importance of the plasma membrane in sensing mechanical stress is well described in animal systems. Studies in *Arabidopsis* have led to the identification of different plasma membrane-localized mechanosensitive ion channels, including proteins that belong to the MSL family, MID1-COMPLEMENTING ACTIVITY1 (MCA1) and its homolog MCA2 [46]. MCA1 was originally characterized by its ability to rescue a MID1 deficient yeast strain. MID1 and MCA1 exhibit low homology, but genetic evidence and the fact that roots of mca1-null plants failed to penetrate a harder agar medium from a softer one [47] support that MCA1 can function as a stretch-activated, plasma membrane-localized Ca^2+^ channel [47]. *MCA2* is the only paralog of *MCA1* found in the *Arabidopsis* genome. The amino acid sequence of MCA2 is to 89.4% similar to the sequence of MCA1. Both also share the same structural features. The N-terminal halves of the proteins consist of an EF-hand-like motif and a coiled-coil motif and are putative protein-kinase domains. The C-terminal halves have up to four putative transmembrane domains and a Cysteine-rich domain [48]. Roots of *MCA2* mutants are incapable of taking up Ca^2+^, and the *mca1*/*mca2* double mutant shows a reduced growth phenotype compared to the wild type [46]. The reduced hyperosmolality-induced [Ca^2+^]i increase (OSCA) family in *Arabidopsis* consists of 15 protein members and two of them, OSCA1 and OSCA1.2, are osmosensitive Ca^2+^-permeable cation channels that can be activated by hyperosmotic treatment [49]. Recently, the atomic structure of *Arabidopsis* OSCA1.2 was shown to display a hydrophobic linker that protruded into the lipid bilayer, acting as a potential sensor of osmotic stress, considering that its partial deletion impairs the opening of the channel under hyperosmotic stress [49].

It is also clear that the lipid environment that surrounds mechanosensitive channels has an influence on the current state of a channel. The force-from-lipid principle describes, that the presence or absence of certain phospholipids can regulate the tension of a bilayer such as the plasma membrane. The tension that is put on the membrane depends on the size of both the tail and the headgroup of each phospholipid. Altering the plasma membrane composition around channels might, therefore, cause conformational changes, resulting in the opening or closure of the channel (also reviewed in [50]).

## 5. The Auxin Efflux Carrier PIN1 Re-Localize at the Plasma Membrane in Response to Mechanical Stress

The phytohormone auxin is a key regulator of growth in plants and mediates developmental responses to internal- and external- physical stimuli. Auxin accumulates at specific sites, thereby inducing local developmental processes within a tissue. These developmental processes are very well studied in the shoot apical meristem (SAM) [51]. In *Arabidopsis*, the SAM is organized into three distinct clonal layers (L1-L3) and consists of a central zone containing slowly dividing stem cells, surrounded by a peripheral zone with rapidly dividing cells where organ primordia are continuously initiated [51]. Cell growth and division push certain pluripotent daughter cells from the central zone to the periphery where they are incorporated into organs or stem tissues. Modeling and mathematical calculations showed that mechanical stress is highly anisotropic in a boundary region between the SAM peripheral zone and new organs, where morphogenesis occurs [52]. Auxin acts as a major regulator of the molecular signaling network that is involved in cell differentiation at the periphery and accumulation of this hormone is regulated by the auxin efflux carrier PIN-FORMED 1 (PIN1), thereby controlling organogenesis. PIN1 polarly localizes at the plasma membrane and exhibits dynamic patterns of expression [53]. It was shown in *Arabidopsis* SAM that PIN1 distribution correlates with the principal direction of mechanical stress, indicating that PIN1 and hereby auxin accumulation might be mechanically regulated, but also that the growth-induced mechanical strain in the boundary zone upregulates PIN1 accumulation. This specific accumulation increases local auxin level, thereby promoting organogenesis [51]. Similarly, dissected tomato SAMs of transgenic lines expressing *pPIN1::PIN1-GFP* were submitted to various mechanical strains and auxin accumulation was then analyzed [54]. Modification of turgor pressure (hypoosmotic and hyperosmotic conditions), application of external force (by pressing with a pulled glass rod), and artificial growth induction collectively showed that the amount and intracellular localization of the auxin efflux carrier PIN1 are sensitive to mechanical alterations [54]. Modulation of the plasma membrane properties alone also explained some of the mechanical effects. Ethanol and dimethyl sulfoxide (DMSO) are both known to expand the plasma membrane, decrease the plasma membrane rigidity and tension and enhance endocytosis [55,56]. Ethanol and DMSO both reduced the PIN1 density at the plasma membrane, without causing cell wall strain. Hypoosmotic treatment reversed this effect, indicating that the plasma membrane retained a healthy activity. Interestingly, co-treatment of ethanol or DMSO with mannitol suppressed the hypoosmotic effect on PIN1 [54]. Since internalization of PINs proteins decreased in plasmolyzed *Arabidopsis* cells (inducing a decrease of the tension at the plasma membrane) and because PIN-polarity is thought to be maintained through differential exocytosis-endocytosis on one cell face, it is reasonable to hypothesize that PIN-polarity might be controlled by the plasma membrane mechanical properties. Nevertheless, this question still remains to be tackled [57,58].

Recently, it was shown in the SAM that mechanical stimulation caused transient changes in cytoplasmic Ca^2+^ ion concentration and that transient Ca^2+^ response was required for downstream changes in PIN1 polarity [59]. Dissected *pPIN1::PIN1-GFP* transgenic inflorescence SAM were treated with LaCl_3_, a plasma membrane Ca^2+^ channel blocker [60]. Treatment with LaCl_3_ blocked the formation of new flower primordia, while the PIN1-GFP localization-pattern remained unchanged. Changing the mechanical pattern in the SAM by performing cell ablations caused a pattern of PIN1 relocalization. Pre-treatment of SAMs with LaCl_3_ completely blocked PIN1 relocalization, indicating that cytosolic Ca^2+^ is crucial for PIN1 relocalization in response to mechanical stress. In contrast, Ca^2+^ was found to be unnecessary for the response of the main mechanical stress readout currently used: the reorganization of microtubules. This reveals that Ca^2+^ specifically-acts downstream of mechanics to regulate PIN1 polarity response [59].

## 6. Cortical Microtubule Dynamics—A Cellular Response to Mechanical Stress

By definition, a mechanical stress is invisible and has to be either calculated according to the shape of the tissues and the physical property of the material creating and restraining the mechanical stress, or it has to be deduced by analyzing its effect. This second strategy is commonly used in geology where the thickness, shape, and nature of the strata stacked on top of each other are indicators for the action of the geological forces at play (compression, contraction, sinking, outcropping, etc.). One of the major readouts used to deduce mechanical stress in plants is the dynamics of the cortical microtubules (cMT), a highly ordered array beneath the plasma membrane. The orientation of the cMTs depends on the local stress pattern and has been correlated to the orientation of cellulose microfibrils. A classical model to explain this correlation postulates that the movement of the cellulose synthase complex (CESA), which is localized at the plasma membrane, is guided by cMTs [8]. The CESA is linked with the cMTs via the CELLULOSE SYNTHASE INTERACTIVE 1 (CSI1) protein. This protein directly binds to CESA and cMTs and was shown to be critical for the co-alignment of CESA with cMTs [61,62,63]. These complexes influence the orientation and the deposition of cellulose microfibrils, which controls the direction of maximal stiffness in cell walls and therefore modifies the shape of the cell. This, on the other hand, changes the local stress pattern towards the neighboring cells, thereby creating a mechanical feedback loop, as these cells need to adapt, which again results in changes of the local stress pattern [64]. Plasma membrane localization of the CESA is also organized by the actin cytoskeleton. It was shown that actin regulates the cellulose synthase lifetime and delivery rate to the plasma membrane. Quantitative image analyses also revealed that actin organization affects cellulose synthase tracking behavior at the plasma membrane and that small compartments were associated with the actin cytoskeleton. The movement of the cellulose synthase is most likely driven by cellulose synthesis and anchorage of cellulose microfibrils in the wall [65].

The dynamics of cMT in accordance to stress pattern can be tested experimentally. The stress pattern of a tissue can be modified by cell ablation (using a needle or a laser), thereby driving the cMT to reorganize in parallel to the new stress pattern [8]. An upcoming theory proposes that the cMTs might act as tension sensors themselves (reviewed in [66]).

In plants, cMTs exhibit dynamic instability, with repeating phases of polymerization (i.e., growth) and disassembly (i.e., shrinkage) separated by pauses, catastrophes, and rescues. cMT dynamics depend on elongation at the plus ends, by continuously repeating catastrophe, rescue events and depolymerization events at their minus ends, and nucleate mainly at the plasma membrane [67]. CMT severing depends on KATANIN (KTN1) as *ktn1* mutants show dramatically reduced frequencies of cMT severing, weakened co-alignment of cMTs and delayed rearrangements. KTN1 activity depends on ROP-interactive CRIB motif-containing protein 1 (RIC1), which is an effector of RHO OF PLANTS 6 (ROP6) GTPase and promotes KTN1 dependent parallel ordering of cMTs [67]. KTN1, RIC1, and ROP6 form a complex, thereby catalyzing the severing and reorganization of cMTs. This complex was also shown to be important for primordium initiation in the SAM [68]. There is evidence that ROP6 is an important anchor for cMT to the plasma membrane [69]. Anchoring of cMTs to the plasma membrane and the signaling pathway behind it remains poorly understood. Analysis of the ROP6 protein revealed a polybasic region adjacent to its C-terminal end. Such polybasic regions have been described to bind to anionic phospholipids [70]. Substitution of seven lysine residues into neutrally charged glutamine residues in the polybasic region of ROP6 abolished in vitro interactions with all anionic lipids [71]. *In planta*, deleting the positive charges of ROP6 C-terminal end or the net negative charge of the plasma membrane caused a ROP6 mislocalization into intracellular compartments. ROP6 localization depends on the presence of phosphatidylserine membrane domains at the plasma membrane, an anionic lipid consisting of two acyl chains that can vary among cell types, which is present in the inner- and outer-leaflets of the lipid bilayer [71,72]. In root tip epidermal cells, ROP6 is immobilized in plasma membrane domains upon activation by auxin. Total Internal Reflection Fluorescence Microscopy (TIRFM) coupled with Fluorescence Recovery After Photobleaching (FRAP) and Photo-Activated Localization Microscopy (PALM) were used to analyze at the single-particle level the dynamics of ROP6 and phosphatidylserine-specific biosensors at the plasma membrane [71]. Phosphatidylserine seems to be necessary for both ROP6 stabilization into membrane domains and signaling. Moreover, phosphatidylserine itself appeared to be present and immobile in these specific membrane domains, suggesting that phosphatidylserine-containing nanoclusters are necessary for ROP6 stability and constitute the functional signaling unit of this GTPase [71]. This opens up the question if a similar regulatory pathway is present for the ROP6-dependent reorganization of cMT. Unfortunately, there is no direct link shown for now. Another cMT-organizing ROP is the ROP11 GTPase. In xylem cells it is locally activated at the plasma membrane and recruits the microtubule-binding protein MIDD1. MIDD1 recruits Kinesin-13A, an ATP-dependent cMT depolymerizing kinesin, to induce depolymerization of cMTs [73,74]. As far as we know, plants have 11 ROPs that can respond to a wide range of signals but their organization is still a black spot [75]. All ROP proteins have a polybasic region at their C-terminal end, and phosphatidylserine, or other anionic lipids, could therefore eventually regulate other members of this family [71].

Cortical microtubules are highly dynamic and present in distinct orientations in the SAM. In the boundary between the newly forming organs and the peripheral zone of the meristem, the tissue is folded and mechanical stress is predicted to be highly anisotropic along the axis of the boundary [8]. It was recently shown, using specific biosensors, that two phosphoinositide phosphates, phosphatidylinositol-4-phosphate (PI4P) and phosphatidylinositol-4,5-bisphosphate (PI(4,5)P_2_), accumulate in the boundary zone of the SAM [76]. Modification of the mechanical stress by tissue ablations and pharmacological approaches induce PI(4,5)P_2_ ectopic accumulation that matches the predicted mechanical stress pattern [76]. These results strongly suggest a relation between phosphoinositide phosphates and cMTs and open the path to explore the role of signaling lipids in mechano-transduction.

## 7. The Plasma Membrane—A Source for Signaling Molecules to Respond to Mechanical Stress

In addition to its regulatory effects, the plasma membrane is also the source for many precursors of different soluble secondary messengers. For example, the classic second messenger inositol triphosphate (IP3) is produced through hydrolysis of PI(4,5)P_2_, a phospholipid that is in plants mainly present in the plasma membrane [15], by membrane-bound PHOSPHOLIPASE Cs (PLCs). PLCs are thought to be activated by cell surface receptors in plants (as it is the case in animals), but they seem to be able to recognize Ca^2+^, suggesting that they are regulated by stimulus triggered Ca^2+^ [77]. IP3 has an important role in releasing intracellular calcium stores in animal cells (mainly from the endoplasmic reticulum) and was described to be important in many regulatory pathways, whereas in plants, the role of IP3 has not been as clearly established [78]. IP3 is involved in the closure of stomata and IP3 level are upregulated in response to different stress conditions [78,79]. High-temperature results in a rapid upregulation of IP3 inducing the increase in intracellular Ca^2+^, which seems to be specific for PLC3 and PLC9 activity [78]. A similar pathway might also be involved in the rapid cytoplasmic increase of Ca^2+^ in response to mechanical stress. In plants, Ca^2+^ signaling was shown to be very important to respond to mechanical stress [59]. Therefore, PLC activity at the plasma membrane might be stimulated by Ca^2+^ or cell-surface sensors resulting in the increase of IP3, which stimulates the release of Ca^2+^ from the endoplasmic reticulum resulting in a cytoplasmic Ca^2+^ spike triggering further cellular responses (Figure 1).

The dynamics of the plasma membrane in response to mechanical stress is still a black spot in plants. In the SAM, PI(4)P and PI(4,5)P_2_ distribution at the plasma membrane correlates with the predicted mechanical stress pattern (and thereby cMT organization) [76]. Nevertheless, the exact contribution of anionic phospholipids to cMT distribution or the cell wall integrity pathways still needs to be shown. New tools, such as Förster resonance energy transfer- (FRET-) based phospholipid specific biosensors (i.e., the recently published phosphatidic acid (PA) specific sensor PAleon [80]) and super-resolution microscopy will help to give us further insights into the function of those lipids on a tissue- and a cellular-level in response to mechanical stress. PA is an important signaling molecule in plants, that is mainly present in the plasma membrane and was shown to be upregulated in response to biotic and abiotic stresses (i.e., osmotic-, heat-, cold-, freeze-stress or after infection with the *Pseudomonas* effector protein AvrRpm1) [81,82]. The synthesis of PA mainly relies on three pathways, the dephosphorylation of diacylglycerolpyrophosphate by lipid phosphate phosphatase, the phosphorylation of diacylglycerol by the diacylglycerolkinases and the cleavage of phosphatidylcholine and/or phosphatidylethanolmine by the PHOSPHOLIPASE D (PLD) protein family [82]. It was also shown that PA plays an important role in microtubule bundling in response to salt stress [83]. A current model predicts, based on genetic and pharmaceutical approaches, that an increase in PLDα1-dependent PA levels at the plasma membrane directly activates the microtubule bundling protein MICROTUBULE-ASSOCIATED PROTEIN 65-1 (MAP65-1), a protein that was shown to directly bind to PA, in response to salt stress. Treatment with high concentration of 1-Butanol, a primary alcohol that can be used as an alternative substrate of the PLDs to decrease PA levels, alone already downregulates cMT bundling, further illustrating that PA might play an important role in microtubule dynamics [83,84]. If this is also true in response to mechanical stress and how this might be regulated on a genetic level is a subject for future studies.

A phospholipase D (PLD) dependent synthesis of phosphatidic acid (PA) was shown to play an important role in cortical microtubule (cMT) bundling in response to salt stress. Phosphatidyl ethanolamine (PE) and phosphatidyl choline (PC) gets hydrolyzed by PLDα1 to PA, hereby activating the MICROTUBULE-ASSOCIATED PROTEIN 65-1 (MAP65-1). A similar mechanism might also be involved in response to mechanical stress. The PLD protein family consist in Arabidopsis of 12 members, that can be classified by their N-terminal domains into the Ca^2+^ activated C2-PLDs and the PX/PH-PLDs [85]. Opening of mechanosensitive Ca^2+^ channels might lead to the activation of C2-PLDs, hereby regulating cMT dynamics. A comparable system could also be involved in the activation of the phospholipase C (PLC) family, as earlier studies indicate that the PLCs might be activated by Ca^2+^ in plants. It was also shown that PLCs are activated by cell surface receptors in mammalian cells. A similar mechanism is thought to also be involved in plants. The PLCs hydrolyze PI(4,5)P2 to inositol triphosphate (IP3). IP3 is a known signaling molecule, that can be further phosphorylated to inositol pyrophosphates, which can fulfill different functions as signaling molecules [86]. In animals, IP3 is recognized by IP3-receptors at the endoplasmic reticulum leading to an intracellular Ca^2+^ spike. Earlier studies indicate, that there might be a similar system in plants.

## 8. Conclusions and Future Perspectives

Understanding how cells and tissues perceive and respond to mechanical stress is a central question to better understand plant development but also responses to biotic and abiotic stress. Indeed, plant development is the result of three essential processes: cell expansive growth, cell division and cellular differentiation, and these three phenomena were described and explained according to the laws of physics [87,88]). Molecular players involved in polar growth of root hairs and pollen tubes are well-described but recent studies revisited these models using mechanics as a new angle to understand how root hairs or pollen tubes are able to physically invade the soil or the stigmatic papillae, respectively [89,90,91]. Similarly, knowledge on biotic and abiotic stresses can be revisited when considering cell mechanics. Abiotic stresses such as a drought- or a salt-stress are inducing a variation of the turgor pressure that will modify cell mechanics [92].

To understand in plants how mechanical stresses are perceived and transduced, it is necessary to identify and characterize more players involved in this signaling pathway. Indeed, one of the readouts used to visualize *in vivo* mechanical stresses is the dynamic reorganization of cMTs [5,61]. MTs and their array anisotropy are visualized using well-established fluorescent fusion proteins that either mark tubulin monomers or decorate microtubules (e.g., tagged versions of microtubule-associated protein (MAP) or microtubule-binding domain (MBD)). Nevertheless, reorganization of cMTs can only be properly analyzed 1–2 h in complex tissues after modification of the mechanical stress, even if they should start to respond immediately after the mechanical perturbation [61]. Moreover, fusion proteins that mark tubulin monomers or decorate microtubules are known to destabilize or stabilize microtubules, respectively. Using tools that perturb the dynamics of one the main actors involved in the studied signaling pathway is evidently problematic.

Many individuals signaling pathways have been proposed or described to play a role in response to mechanical stress (Figure 2). But for now, the connection between signaling pathways involved in mechanical stress perception and transduction is still poorly understood. Finally, how mechanical stresses are used by cells or tissues to modify their shape that will generate a new mechanical stress, that would directly link mechanical induced Ca^2+^ signaling, the re-distribution of PIN1 (and therefore auxin), ROPs localization and activity and cMT dynamics. Future studies will hopefully reveal such a network.

## Figures and Tables

**Figure 1 plants-09-00505-f001:**
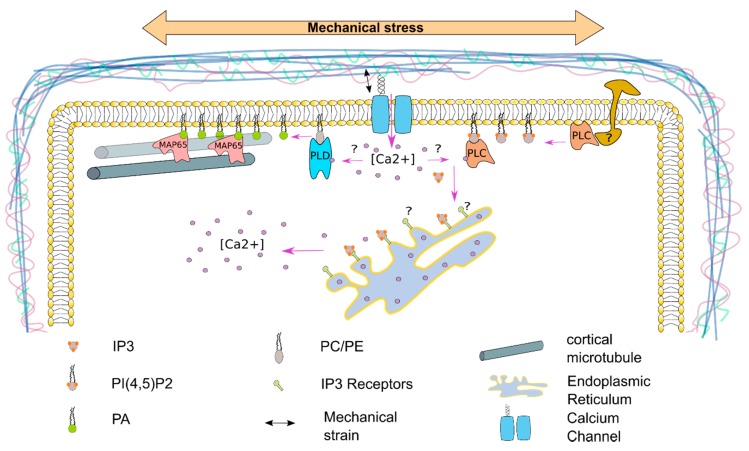
A putative network for plasma membrane derived signaling molecules.

**Figure 2 plants-09-00505-f002:**
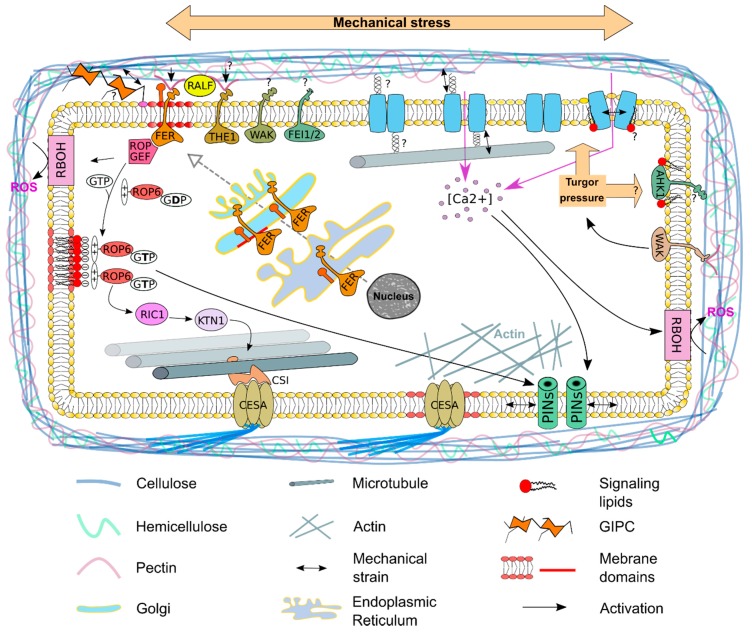
Role of the plasma membrane in mechano-sensing. Mechanical stress can modify the composition and/or organization of the cell wall. This may be perceived by transmembrane receptor kinases such as FERONIA (FER), THESEUS1 (THE1), WALL-ASSOCIATED KINASE (WAK) and FEI1 - FEI2 receptor-like kinases. Localization of FER in membrane domains may be done thanks to the interaction with LLG1, a glycosyl-phosphatidylinositol-anchored protein synthetized in the endoplasmic reticulum and localized in membrane domains in the Golgi. Selective opening of stretch-activated channels can open when a mechanical tension is pulling them (this implies they have to be anchored outside and/or outside the cell by an unknown mechanism represented by a spring) or by a modification of the membrane tension, that may be regulated by the membrane curvature and a specific lipid composition of the plasma membrane. Another hypothesis is that long decoration present in complex lipids such as Glycosyl Inositol Phospho Ceramides (GIPC) that can be formed by up to 14 sugars will interact with the cell wall and create a physical link between the lipid bilayer and the cell wall. Turgor pressure may potentially be perceived by ARABIDOPSIS HISTIDINE KINASEs (AHK) 1. Channel opening leads to an increase of calcium concentration ([Ca^2+^]) that activates respiratory burst oxidase homologues (RBOH) that will produce reactive oxygen species (ROS). Ca^2+^ is also needed for PIN1 relocalization in response to mechanical stress. Accumulation at the plasma membrane of RHO OF PLANT (ROP) 6 is activated by ROP-GEF that interacted with FER. ROP6 localization into membrane domains is driven by positive stretch that interact with the negatively charged signaling lipids. ROP6 can activate ROP-interactive CRIB motif-containing protein 1 (RIC1) and promotes KATANIN1 (KTN1) dependent parallel ordering of cMT. Cellulose synthase interactive protein 1 (CSI1) acts as a linker protein between CESA complexes and microtubules. Plasma membrane localization of the CESA and PINs is at least partially regulated by actin cytoskeleton.

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
