# Peer review of "The Plasma Membrane—An Integrating Compartment for Mechano-Signaling"

_plants, 2020, doi:10.3390/plants9040505_

Round 1

Reviewer 1 Report

This review aimed to deepen the knowledge on the role of the cell membrane in mechano-sensing in a satisfactory way. The review is extensive and will serve as a good source of information on the subject. I only have some minor observations that are listed in the following lines.

  1. How was the literature for this review chosen? clarify at the end of the introduction.
  2. Rewrite the conclusion. Avoid using illustrations and references in the conclusion. The conclusion should not be a place to present new facts.
  3. Add a list of abbreviations.
  4. Add graphic illustrations to the text to make it easier for the reader to follow. 

Author Response

This review aimed to deepen the knowledge on the role of the cell membrane in mechano-sensing in a satisfactory way. The review is extensive and will serve as a good source of information on the subject. I only have some minor observations that are listed in the following lines.

  • How was the literature for this review chosen? clarify at the end of the

We agree with this comment. We clarified how we choose the literature at the end of the introduction. We highlighted that we made a focus on the plasma membrane and mechanosensing.

  • Rewrite the conclusion. Avoid using illustrations and references in the The conclusion should not be a place to present new facts.

We completely agree with this comment. We moved most of the conclusion to a new section where we present the plasma membrane as a reservoir for secondary messenger that can be used in response to a mechanical stress. We open the field by explaining that many studies can be revisited by considering the cell mechanics.

  • Add a list of abbreviations.

We added a list of abbreviation at the end of the manuscript.

  • Add graphic illustrations to the text to make it easier for the reader to follow. 

This was a good suggestion. We added one figure to illustrate what is, in my point of view, the missing point to understand signal internalization in general: what is the role of the lipid composition of the plasma membrane to activate, regulate and tune signalling pathways.

Reviewer 2 Report

The manuscript entitled “The Plasma Membrane – An Integrating Compartment for Mechano-Signaling “ by Frank Ackermann and Thomas Stanislas presents a comprehensive review of the current knowledge about the role of plasma membrane in mechanical stress perceiving and the signal transduction into the cell.

The manuscript is written in satisfactory manner, with a deep insight into already published high quality papers, merging different aspects of the topic. The provided scheme presents a nice compilation of derived conclusions.

Author Response

We thank reviewer 2 for his kind review.

Reviewer 3 Report

This review addresses a very interesting and central question in the field of plant cell biology: How do plant cells perceive forces? In their review, the authors focus on the role of the plasma membrane and provide a rich and complete overview of the current state of knowledge.

For my taste, the authors overemphasize (in the introduction) that the role and importance of the plasma membrane in the perception of forces have been "overlooked" in the past. In fact, until a few years ago, the cell wall enjoyed much easier experimental access than the plasma membrane. This brings me to my only (and general) point of criticism: Which methods were used in the past? Which ones are available today? And which experiments could be decisive in finding the answer to the question of how plant cells perceive forces.

I list my detailed comments below:

  • smaller typos in lines 28, 31, 172, 180 and many more
  • line 32: "major changes in structure" ... Could the authors be more precise? Structure of what?
  • line 68: "was considered as a simple barrier" ... Is there a reference for this statement?
  • lines 83-85: "revealed the existence of particular domains within the plasma membrane called membrane raft" ... How were rafts shown to exist in plants? What methods were used?
  • lines 87-89: Is there a reference for this statement/sentence?
  • lines 108-110: To my knowledge, it has only been shown that FER binds to demythHG in vitro, not in vivo.
  • lines 127-128: How was this shown? Which method was used?
  • lines 132-134: This is a very long and unclear conclusion sentence. Please revise.
  • lines 135-138: How was this shown? Which method was used? What exactly is the evidence?
  • lines 168-176: The paragraph is missing a clear statement on how all this was measured. Please add more detail which methods were used.
  • lines 197-199: How was this measured?

I think that the "Conclusion and Future Perspectives" section could be written more efficiently by focussing better on how exactly the big scientific question can be approached using what we already know. Currently, this section is more of a potpourri of scientific snippets.

Author Response

This review addresses a very interesting and central question in the field of plant cell biology: How do plant cells perceive forces? In their review, the authors focus on the role of the plasma membrane and provide a rich and complete overview of the current state of knowledge.

For my taste, the authors overemphasize (in the introduction) that the role and importance of the plasma membrane in the perception of forces have been "overlooked" in the past. In fact, until a few years ago, the cell wall enjoyed much easier experimental access than the plasma membrane. This brings me to my only (and general) point of criticism: Which methods were used in the past? Which ones are available today? And which experiments could be decisive in finding the answer to the question of how plant cells perceive forces.

Thank for this general comment. We modified several parts in the text to state the clear importance of the cell wall as a first layer and to highlight that player involved to perceive the state of the cell wall are located at the plasma membrane.

I list my detailed comments below:

  • smaller typos in lines 28, 31, 172, 180 and many more

We agree with this comment. We asked help from native English speaker to proofread the paper.

  • line 32: "major changes in structure" ... Could the authors be more precise? Structure of what?

We modified the sentence to be more precise from “The shape changes, both at the cell and tissue level, rely on developmental-related gene expression and on the concentration and polarity of growth factors that induce major changes in structure, which by definition depend on the laws of mechanics.” to “The shape changes, both at the cell and tissue levels, rely on development-related gene expression and on the concentration and polarity of growth factors that induce major changes in the internal pressure and the property of extra-cellular matrix, which by definition depend on the laws of mechanics.”

  • line 68: "was considered as a simple barrier" ... Is there a reference for this statement?

We cited the review “Gronnier, J.; Gerbeau-Pissot, P.; Germain, V.; Mongrand, S.; Simon-Plas, F. Divide and Rule: Plant Plasma Membrane Organization. Trends Plant Sci 2018, 23, 899-917, doi:10.1016/j.tplants.2018.07.007. In this review, authors described the evolution of the plasma membrane concept, from a simple barrier to a dynamic and complex signalling compartment.

  • lines 83-85: "revealed the existence of particular domains within the plasma membrane called membrane raft" ... How were rafts shown to exist in plants? What methods were used?

We added this paragraph to explain how raft were first discover in plant: “ In plants, the existence of membrane rafts was initially suggested by the identification of a low-density TritonX-100 insoluble fraction [11]. Possible artifacts generated by the use of detergents at low temperatures keep membrane raft concept under active debate. Finally, in situ characterization of plasma membrane organization by immunoelectron microscopy and super-resolution microscopy revealed the presence of <100 nm clusters on the PM of Arabidopsis (review in [11]).”

  • lines 87-89: Is there a reference for this statement/sentence?

The reference for this sentence was: 17.  Schrick, K.; DeBolt, S.; Bulone, V. Deciphering the molecular functions of sterols in cellulose biosynthesis. Front Plant Sci 2012, 3, doi:ARTN 84. Thank you for your remark, we added the reference in the text.

  • lines 108-110: To my knowledge, it has only been shown that FER binds to demythHG in vitro, not in vivo.

We agree with the remark. Fer was shown to interact with demythHG in vitro only. This was a mistake in the text. We modified the sentence accordingly.

  • lines 127-128: How was this shown? Which method was used?

We revised this entire paragraph since it was not clear how we came to the hypothesis that GIPc could be sensor or the CW. We modified this original paragraph: “Lipids are also potential sensor of the cell wall. Glycosyl Inositol Phospho Ceramides (GIPCs) are organized in membrane domains at the plasma membrane acting as signaling molecules. GIPCs are structured in a trihydroxylated LCB moiety amidified by a 2-hydroxylated Very Long Chain Fatty Acid linked to an inositol–glucuronic acid and can be further modified by adding different saccharides. It was shown that GIPCs are capable of binding rhamnogalacturonan II, which is a fraction of pectin (reviewed in [25]). This makes the GIPCs to a putative link of the plasma membrane and the cell wall, which could access the plasma membrane to recognize the current physical state of the cell.” With this new paragraph: ” Lipids are also potential direct sensors of the cell wall. Glycosyl Inositol Phospho Ceramides (GIPCs) are structured in a trihydroxylated LCB moiety amidified by a 2-hydroxylated Very Long Chain Fatty Acid linked to an inositol–glucuronic acid and can be further modified by adding different saccharides to the polar head. It was shown that GIPCs are capable of binding rhamnogalacturonan II, which is a fraction of pectin [25]. GIPCs are lipids with very long saturated acyl chains organized in membrane domains at the plasma membrane [26]. It was calculated that the carbon atom acyl chains of GIPC are too long to fit in one leaflet and will therefor penetrate within the inner leaflet and interdigitate with the inner leaflet lipid acyl chains [26]. This makes GIPCs putative links between the plasma membrane and the cell wall, which could access the plasma membrane to recognize the current physical state of the cell.”

  • lines 132-134: This is a very long and unclear conclusion sentence. Please revise.

We revised this entire paragraph since it was not clear how we came to the hypothesis that GIPc could be sensor or the CW. We modified this original paragraph: “Lipids are also potential sensor of the cell wall. Glycosyl Inositol Phospho Ceramides (GIPCs) are organized in membrane domains at the plasma membrane acting as signaling molecules. GIPCs are structured in a trihydroxylated LCB moiety amidified by a 2-hydroxylated Very Long Chain Fatty Acid linked to an inositol–glucuronic acid and can be further modified by adding different saccharides. It was shown that GIPCs are capable of binding rhamnogalacturonan II, which is a fraction of pectin (reviewed in [25]). This makes the GIPCs to a putative link of the plasma membrane and the cell wall, which could access the plasma membrane to recognize the current physical state of the cell.” With this new paragraph: ” Lipids are also potential direct sensors of the cell wall. Glycosyl Inositol Phospho Ceramides (GIPCs) are structured in a trihydroxylated LCB moiety amidified by a 2-hydroxylated Very Long Chain Fatty Acid linked to an inositol–glucuronic acid and can be further modified by adding different saccharides to the polar head. It was shown that GIPCs are capable of binding rhamnogalacturonan II, which is a fraction of pectin [25]. GIPCs are lipids with very long saturated acyl chains organized in membrane domains at the plasma membrane [26]. It was calculated that the carbon atom acyl chains of GIPC are too long to fit in one leaflet and will therefor penetrate within the inner leaflet and interdigitate with the inner leaflet lipid acyl chains [26]. This makes GIPCs putative links between the plasma membrane and the cell wall, which could access the plasma membrane to recognize the current physical state of the cell.”

  • lines 135-138: How was this shown? Which method was used? What exactly is the evidence?

We modified this paragraph to make it clearer. “So far, no turgor pressure rheostat has clearly been characterized in Arabidopsis. There is evidence suggesting that WALL-ASSOCIATED-KINASES may provide a molecular mechanism linking cell wall sensing and turgor maintenance in growing cells [27]. In vitro, the WAK extracellular domain binds to pectin oligosaccharides in a Ca2+-dependent manner [28] and single wak2 mutation exhibits a dependence on appropriate sugars and salts and reduces vacuolar invertase activity in the roots [27], which in turn is known to be involved in systems of turgor maintenance. Yet there is, for now, no proof for this hypothesis.

  • lines 168-176: The paragraph is missing a clear statement on how all this was measured. Please add more detail which methods were used.

To show that channel can open in response to a mechanical stress, patch clamp technic was used. We added this sentence in the text: “Patch clamp is a powerful technique that allows in a single cell the recording of channel activity with a high resolution in time and ion flux. Using this technique, it was shown that the state of the ion channel depends on the tension force that is applied to the plasma membrane [38] [39]. No membrane tension leads to closed channels. In the presence of tension conformational changes lead to the opening”

  • lines 197-199: How was this measured?

We modified the text to explain how authors hypothesis that MCA1 is mechanosensory

MID1 and MCA1 exhibit low homology, but genetic evidence and the fact that roots of mca1-null plants failed to penetrate a harder agar medium from a softer one [47] support that MCA1 can function as a stretch activated, plasma membrane-localized Ca2+ channel [47].

  • I think that the "Conclusion and Future Perspectives" section could be written more efficiently by focussing better on how exactly the big scientific question can be approached using what we already know. Currently, this section is more of a potpourri of scientific snippets.

We completely agree with this comment. We moved most of the conclusion to a new section where we present the plasma membrane as a reservoir for secondary messenger that can be used in response to a mechanical stress. We open the field by explaining that many studies can be revisited by considering the cell mechanics.